# Cluster Analysis and Discriminant Analysis for Determining Post-Earthquake Road Recovery Patterns [note 1]

**DOI:** 10.3390/s22062213

**Published:** 2022-03-12

**Authors:** Jieling Wu, Mitsugu Saito, Noriaki Endo

**Affiliations:** 1Design and Media Technology, Graduate School of Engineering, Iwate University, Morioka 020-8551, Japan; wujl0612@yahoo.co.jp; 2Regional Innovation and Management, Graduate School of Arts and Sciences, Iwate University, Morioka 020-8550, Japan; endo@iwate-u.ac.jp

**Keywords:** 2011 Tohoku earthquake, big data analysis, cluster analysis, digital road map, discriminant analysis, Fukushima prefecture, geographic information system (GIS), probe-car telematics data, vehicle tracking map

## Abstract

The transport network in eastern Japan was severely damaged by the 2011 Tohoku earthquake. To understand the road recovery conditions after a large earthquake, a large amount of time is needed to collect information on the extent of the damage and road usage. In our previous study, we applied cluster analysis to analyze the data on driving vehicles in Fukushima prefecture to classify the road recovery conditions among municipalities within the first six months after the earthquake. However, the results of the cluster analysis and relevant factors affecting road recovery from that study were not validated. In this study, we proposed a framework for determining post-earthquake road recovery patterns and validated the cluster analysis results by using discriminant analysis and observing them on a map to identify their common characteristics. In addition, our analysis of objective data reflecting regional characteristics showed that the road recovery conditions were similar according to the topography and the importance of roads.

## 1. Introduction

The Tohoku earthquake on 11 March 2011, caused major damage throughout an extensive region of transport routes in eastern Japan. Main roads and railways ceased to function for a long period, and the lives of people affected by the earthquake were significantly affected [1]. Starting the day after the earthquake, the Ministry of Land, Infrastructure, Transport, and Tourism (MLIT) implemented a “road clearance” policy to open up as many roads as possible to vehicular traffic [2]. This operation involved securing rescue and relief routes on many national highways, extending from inland areas toward the Pacific coastal area of Tohoku. The main routes in the Tohoku region were restored within the first week after the serious earthquake at a speed that surprised the whole world.

In Fukushima prefecture, the majority of the coastal region showed a seismic depth of over 6 on the Richter scale, the coast was hit by a large tsunami, and the resulting tremors prompted a sequence of collapses of infrastructure and buildings. Furthermore, the effects of the accident at the Fukushima Daiichi nuclear plant caused by the earthquake were felt throughout the entire world [3].

In the event of a disaster, the collection and consolidation of road information is a time-consuming process for various emergency, rescue, and recovery operations. Therefore, a vehicle tracking map was built by Honda Motor Company, Ltd., Japan to quickly determine the road conditions after the 2007 Niigata-ken Chuetsu-oki earthquake [4]. This system can be used to obtain reference information to support evacuation and rescue operations in disaster areas based on actual vehicular traffic data, showing the traffic routes that are accessible after a major earthquake. After the Tohoku earthquake, on 19 March, ITS (Intelligent Transport Systems) Japan began to integrate probe-car telematics data from private automakers (e.g., Honda, Toyota) to provide information on traffic records and road closures in the affected areas [5].

In the field of post-earthquake road recovery research, there are many research reports on the recovery of motorways and general national roads after the 2011 Tohoku Earthquake [6,7]. However, there are few reports on municipal road recovery related to the daily lives of residents. In previous related studies [8,9,10,11,12,13], the G-BOOK telematics data of the Toyota Motor Corporation were used to survey road recovery after the Tohoku earthquake. The affected prefectures in the Tohoku region were divided into several large areas, and the authors concluded that road restoration varied between these areas. In two of our previous studies [14,15], the same vehicular driving data from the Toyota Motor Corporation in the Fukushima prefecture were divided into seven regions, and it was found that in the six months following the 2011 Tohoku earthquake, the speed of road use recovery in inland areas was slower than that in coastal areas. We concluded that the recovery of roads was much slower in areas that were narrow, steep-walled, and mountainous. In addition, studies [14,15] compared regions in different prefectures, coastal and inland, that reached 90% recovery rates. These areas had similar recovery dates, which illustrate similar rates of recovery between regions. However, the recovery in these seven regions was affected by local consensus [16], which we believe was caused by broad classification, road restoration speed differs between municipalities in the same region. Moreover, the similarity of road restoration between regions should not be seen in terms of similarity at one time alone, but should instead be considered in terms of the similarity of the entire restoration process. Hence, we investigated municipalities in Fukushima prefecture using cluster analysis to classify municipalities with similar road recovery rates [17]. Additionally, we visualized cluster analysis results on a map and observed that road restoration in municipalities was related to the geographical location and topography. The study concluded with the same cluster analysis method used in Miyagi prefecture and was visualized on a map to draw conclusions related to the topography. However, this study did not validate the classification results after obtaining the cluster analysis results regarding road restoration. In a related study [18], we analyzed road recovery in the Fukushima prefecture regarding not only the geographical location and topography but also the population density. In addition, we divided each cluster into four zones and used road closure information to verify the results of the road use recovery. With one exception, the order of these zones in terms of road use recovery was the same as that of road closures being lifted. The cluster analysis results of road recovery in the Fukushima prefecture have not yet been fully validated. Moreover, the visualization on a map has not been tested with objective data to draw conclusions related to geographical location, topography, and population density. We wanted to explore other factors, besides these three, that influence road restoration. Hence, we believe that further detailed analysis is needed.

The purpose of the study was to determine the recovery patterns of post-earthquake local roads using objective data to support disaster mitigation measures. We targeted the municipal road network, which is one of the most essential elements for rescuing victims and supplying them with daily commodities, and surveyed the conditions and recovery patterns of roads accessible to motor vehicles in the municipalities of Fukushima prefecture in the first six months after the disaster. To this end, we proposed the following framework (Figure 1). First, the vehicular driving data were processed to derive the recovery conditions of each municipality at each time period in the first six months after the earthquake. Then, cluster analysis and discriminant analysis were used to identify clusters with similar road recovery rates. Finally, the clusters were observed on a map using GIS to detect their common characteristics and verify them with objective data.

## 2. Materials and Methods

### 2.1. Vehicle Tracking Map

The vehicle tracking map (Figure 2) was built from the G-BOOK telematics data from the Toyota Motor Corporation, which has been available on the internet since 18 March 2011 following the 2011 Tohoku earthquake [19]. Toyota is the largest car manufacturer in Japan, and its vehicle driving data reflect the road conditions after the Tohoku earthquake. The data used in this study were collected in 54 municipalities in the Fukushima prefecture (Figure 3) between 18 March and 30 September 2011 (i.e., approximately six months following the 2011 Tohoku earthquake), excluding municipalities located in the no-go zone due to the accident at the Fukushima Daiichi nuclear power plant.

### 2.2. System

#### 2.2.1. Hardware

The computations were performed on a standard PC laptop with a Core i7–10510U CPU (1.8 GHz) and 16 GB memory (ASUS Expert Book B9450FA: Taiwan).

#### 2.2.2. Software

This study used QGIS version 2.18.20 [20], IBM SPSS Statistics 23 [21], and Microsoft Excel 2019 software running on the Windows 10 Professional operating system.

### 2.3. Data Processing

(1)The vehicle tracking maps constructed from the G-BOOK telematics data were provided in the Google Maps KMZ format. For our analysis, we first converted the KMZ files into SHP files (i.e., shape files), which are compatible with ArcGIS using the “ogr2ogr” function [22] on the Linux operating system [23].The data coordinates were converted from the terrestrial latitude and longitude into the x and y coordinates in a rectangular coordinate system.(2)After merging the daily data into weekly data and removing duplicates, we were able to calculate the exact available road distance for a given week.(3)Next, we calculated the proportion of the cumulative distance up to the specified date and considered the cumulative distance up to 30 September 2011, to be 100%.

### 2.4. Cluster Analysis

Cluster analysis is the task of clustering a set of objects such that all objects in a cluster are similar to one another and at the same time are distinctly different from all objects outside of this cluster. It is a major task of exploratory data analysis, in which observations are divided into meaningful groups whose members share common characteristics. It is a common technique for statistical data analysis and is used in many fields, including pattern recognition, image analysis, information retrieval, bioinformatics, data compression, computer graphics, and machine learning [24].

Using this method, we decided to classify all municipalities in Fukushima prefecture using cluster analysis to determine the common characteristics shared by municipalities with similar road recovery conditions.

There are various methods of cluster analysis, of which k-means clustering and hierarchical clustering analysis are more commonly used. K-means clustering requires the number of clusters to be specified in advance and is suitable for large data, while hierarchical clustering analysis determines the number of clusters based on the output results and is suitable for small data types. As our analysis sample is only 54 municipalities with small data, we chose unsupervised hierarchical clustering analysis.

The basic logic of hierarchical clustering analysis is that each case (or variable) is first considered as a class, then grouped into smaller classes based on the distance or similarity between the cases (or variables), and then gradually grouped upwards based on the distance or similarity between the classes, until all the cases are aggregated into one large class.

In the hierarchical cluster analysis, we employed Ward’s method [25], which is also the most commonly used. As a procedure for grouping similar objects, Ward’s method aims to minimize the sum of squared errors between two groups for all variables.

The squared Euclidean distance between each pair of observations is used to measure the similarity between groups, with shorter distances indicating greater similarity. If there are n attribute variables measuring the “distance” and the “distance” between No. j case and No. k case, the squared Euclidean distance can be expressed by the Equation (1):(1)$$e_{jk}=\overset{n}{\underset{i=1}{\sum }}{\left(X_{ij}-X_{jk}\right)}^{2}$$

Using the cumulative data from Section 2.3, we obtained the percentage of road use recovery in each municipality. Then, we introduced the percentages into SPSS Statistics software and used Ward’s method with the squared Euclidean distance as the measurement interval in hierarchical cluster analysis to obtain the cluster analysis results. The number of clusters was chosen according to the stopping rule (a large percentage drop in the agglomeration coefficients followed by a plateau). The results were also confirmed as seven clusters (Table 1) by visual inspection of the dendrogram.

### 2.5. Discriminant Analysis for Validation of the Cluster Analysis Results

#### 2.5.1. Canonical Discriminant Analysis

Canonical discriminant analysis is a classification model that works by identifying a projection hyper plane in k-dimensional space such that the projections of the same categories on that hyper plane are as close as possible to each other while the projections of different categories are as far apart as possible. If the results obtained from the cluster analysis can be fitted with the discriminant analysis equation, this classification result is considered valid [26].

#### 2.5.2. Canonical Discriminant Function Determination

Therefore, we used the number of clusters as the dependent variable and the date of recovery as the independent variable and chose “enter independent together” for the discriminant analysis in the SPSS statistics software. The larger the eigenvalue is, the better the linear discriminant function obtained. According to Table 2, the canonical correlations of the first two functions derived from SPSS both reach 85% or more, with the two functions together explaining 86.3% of the variance. Furthermore, the closer the Wilks’ lambda value is to 0, the better the group is identified, and the significance of the first two functions was 0.000 in Wilks’ lambda test (Table 3). Therefore, we believe that the results of the cluster analysis are successfully captured by using the first two functions.

## 3. Results

### 3.1. The Cluster Analysis Results

Municipalities with similar road recoveries were divided into seven clusters according to the results of the cluster analysis (Table 1).

The order of the date at which the recovery reached 90%, averaged for each cluster, is 3 > 5 > 1, 4 > 6 > 2 > 7 (Table 4, Figure 4). We displayed the cluster of each municipality on the map via GIS (0 is a closed area due to the Fukushima Daiichi nuclear power plant accident, Figure 5).

Consider the location (Figure 3 and Figure 6) and the recovery order of each cluster:

Cluster 3: Municipalities located adjacent to Koriyama, the largest city in the Fu-kushima prefecture: the roads in the Nakadori Basin were generally the fastest to recover.

Cluster 5: Roads located in basins, lowlands, and large cities: the speed of road recovery was the second-fastest here. The roads in the municipality here are relatively dense, and the population density is high.

Cluster 1: In municipalities on the west side (Echigo Mountains) due to snow and on the east side (coastal lowlands) due to tsunamis, roads gradually recovered after the disaster road closure was lifted.

Cluster 4: In municipalities located in the mountains and the basin, urban locations are relatively populous, and the speed of road recovery was moderate compared to other clusters.

Cluster 6: Recovery was slow here due to mountains (Echigo Mountains) and snow. Similar to Cluster 2, they were affected by lingering snow, but due to their location by the Ban-Etsu Expressway, road use recovery tended to be faster than in Cluster 2.

Cluster 2: Municipalities here were concentrated in the mountains (Ou Mountains, Abukuma Highlands), and road recovery was slow. Additionally, there are few major arterial roads.

Cluster 7: This cluster had the slowest recovery due to areas of heavy snowfall and mountainous areas (Echigo Mountains). Note that in the inland Tohoku region, especially in the mountainous areas, roads were closed from the previous winter until June this year because of snow [27].

In the disaster areas, similar recovery conditions were observed depending on geographical location, topography, population density, damage, road importance, road density, and snow. Recovery in lowland areas seemed to be faster than in mountainous areas. In general, in Fukushima prefecture, road restoration is concentrated in the middle basin first, and then extended to the sides; the higher the terrain, the slower the recovery.

Thus, the seven road recovery clusters were classified based on their characteristics.

Cluster 3—fast recovery, dense roads, small plain municipalities.

Cluster 5—fast recovery, dense roads, large plain municipalities.

Cluster 1—medium recovery, affected areas, mixed mountain–plain municipalities.

Cluster 4—medium recovery, low population density, mixed mountain–plain municipalities.

Cluster 6—slow recovery, with one main road, mountainous municipalities.

Cluster 2—slow recovery, low road density, mountain and snow municipalities.

Cluster 7—slow recovery, mountain and heavy snow municipalities.

### 3.2. Validated Results of Discriminant Analysis on Classification

#### 3.2.1. Standardized Canonical Discriminant Function

Regarding the standardized canonical discriminant function coefficients, the larger the absolute value of the coefficient is, the greater the contribution to the discriminant function. As shown in Table 5, for Function 1, the order according to the coefficients is Apr—2 w > Apr—1 w > Mar—4 w > May > Jun > Apr—3 w > Apr—4 w > Mar—3 w > Jul > Aug. It is clear that the recovery rates in the 2nd week of April, the 1st week of April, and the 4th week of March contribute the most to the discriminant function; that is, the recovery rates in the 4th, 3rd, and 2nd weeks after the Tohoku earthquake are the most important factor in the classification of the recovery cluster. Similarly, the recovery rates in weeks 2, 3, and 4 of April contribute the most for Function 2.

Pooled within-group correlations between discriminating variables and standardized canonical discriminant functions are shown in Table 6. The variables ordered by the absolute size of correlation within Function 1 are Mar—4 w > Apr—2 w > Apr—1 w > Mar—3 w > Apr—3 w > Apr—4 w > May > Jun > Aug > Jul. Similarly, the order of function 2 is Mar—3 w > Apr—2 w > Apr—1 w > Apr—3 w > Apr—4 w > Jun > May > Aug > Jul > Mar—4 w. We believe this is because the difference in recovery rates between clusters is more pronounced in the first 6 weeks when 30 September is considered to be 100% road recovery. As time progresses, the differences are relatively less pronounced.

Although the correlations with the two functions are shown in Table 6, the variables for Apr—2 w are both more advanced. As the previous results (Table 2) show that the first discriminant function carries most of the discriminant information, this suggests that the variable for Mar—4 w may play a major role in the discriminant analysis. We believe that this is linked to the government’s road recovery policy, namely “road clearance”, prioritizing the restoration of the Tohoku Expressway and major national highways and opening up the roads to the affected areas along the coast. Both Cluster 3 and Cluster 5 are concentrated within the area of roads covered by this policy, so the road recovered relatively quickly.

#### 3.2.2. Unstandardized Canonical Discriminant

From Table 7, we can obtain the unstandardized canonical discriminant functions evaluated as group means. The centroids of the discriminant functions for each cluster are given in Table 8.
F1 = 0.039 × X_1_ + 0.104 × X_2_ − 0.123 × X_3_ + 0.148 × X_4_ + 0.063 × X_5_ − 0.053 × X_6_ − 0.144 × X_7_ + 0.356 × X_8_ − 0.190 × X_9_ − 0.001 × X_10_ − 14.3(2)
F2 = 0.136 × X_1_ + 0.013 × X_2_ + 0.131 × X_3_ − 0.329 × X_4_ + 0.263 × X_5_ − 0.270 × X_6_ + 0.009 × X_7_ − 0.171 × X_8_ + 0.169 × X_9_ + 0.119 × X_10_ − 1.76 (3)

The unstandardized discriminant function and the clustering centers are represented in the following diagram (Figure 7). Based on the use of these two discriminant functions to predict the classification, the correct rate was 92.6% (Table 9). This shows that the results of the cluster analysis can be successfully tested with discriminant functions.

From the results of the discriminant analysis prediction shown in Table 9, it can be seen that Cluster 1 and Cluster 4 each predicted the wrong set for one another. The two discriminant functions are relatively close in the distribution in Figure 7. They are also very close to each other, as observed on the map (Figure 5). Cluster 4 and Cluster 1 border Cluster 3 and Cluster 5, and their road restoration was affected by and completed immediately after that of Cluster 3 and Cluster 5. Additionally, Cluster 4 and Cluster 1 belong to plains and mountainous terrain. Cluster 2 showed one misjudgment as Cluster 4, and Cluster 3 showed one misjudgment as Cluster 5; both of these misjudgments are variables that are distributed very closely in Figure 4. In contrast, Clusters 5, 6, and 7 have a 100% discrimination rate. The topography of the three clusters is well-differentiated, with Cluster 5 on the plains and Clusters 6 and 7 in areas with mountains and heavy snow.

## 4. Discussion

To prove our hypothesis, we collected data reflecting geographic location, topography, population density, damage, road importance, road density, and snow to examine their relationship with road recovery.

For road-related factors, we used the March 2011 version of the Digital Road Map (DRM) [28] to calculate the distance and area of roads in each municipality. DRM was supplied by the Japan Digital Road Map Association. It is the standard national digital roadmap database used to assist Japanese ITS infrastructures. This database consists of virtual cartographic data in which locations and other information are expressed in numeric form so that computer systems can recognize roads, intersections, etc. The DRM Database allows car vehicle systems and the like to display road maps on their displays and find suitable routes to a destination that avoid traffic congestion. The inspiration for the use of DRM is that the latest version of its map data also includes information about road elevation, which can reflect the actual condition of the road topography. However, older versions, such as the 2011 map of the Fukushima prefecture used in our study, do not have additional elevation information, and we decided to add it ourselves to see how road recovery and topography relate specifically. Additionally, the road density and area of each municipality in the Fukushima prefecture in 2011 were calculated using DRM.

### 4.1. Data Collection on Factors Affecting Road Recovery

The individual factors affecting road recovery are described as follows.

#### 4.1.1. Geographic Location and Topography

The 2011 Digital Road Map data in the SHP file only provide the latitude and longitude and are generally saved as x and y attributes in the geometry. We used the “add z value” function of GIS to add the digital elevation model (DEM) data [29] to the z values of the road data.The road data are stored in intervals in the SHP file’s properties. We calculated the average of the z values (i.e., the average elevation) of the roads in intervals and saved them together in the attribute table.We calculated the distance of roads with an average elevation of less than 50 m, 50 to 100 m, 100 to 200 m, 200 to 500 m, and more than 500 m for each municipality and then calculated the percentage of road length at each elevation compared to the total road length of the municipality.

#### 4.1.2. Population Density

Population density can be obtained using census data [30]. Here, we used data from 2010, the year before the Tohoku earthquake, as a reference.

#### 4.1.3. Damage

Earthquake seismicity is often used as a criterion for predicting damage [31]. We collected measured seismic intensities from the Japan Meteorological Agency for each municipality in the Tohoku region for the 2011 Tohoku earthquake [32].

#### 4.1.4. Road Importance

To determine the importance of roads, we calculated the distances of highways and national roads that were given priority for restoration after the disaster as a percentage of the total distance of roads in the municipality.

#### 4.1.5. Road Density

The ratio of road distance to the area in the municipality was also used as one of the indicators affecting road rehabilitation.

#### 4.1.6. Snow

We calculated the road closure rate due to snow based on the information on the winter closure route in the Fukushima prefecture [27].

### 4.2. Pearson Correlation Analysis

We used Pearson correlation analysis to determine the relationship between road restoration patterns and the above factors. Values less than 0.05 indicate a significant correlation and statistical significance. From Table 10, clusters and elevations from 100 to 200 m and above 500 m are all well below 0.05, indicating a significant correlation between them. Other influencing factors besides topography such as road importance, road density, damage, and snow were all significantly correlated with road recovery patterns. Surprisingly, the population density was not significantly correlated with road recovery patterns (Table 11). It shows that the previous study [18] suggested that the correlation between road recovery and population density was not justified.

Similarly, we show the cluster analysis results from Miyagi prefecture from Reference [17] to see how they relate to road elevation. Road rehabilitation patterns in Miyagi prefecture were significantly associated with road elevations below 50 m and from 200 to 500 m (Table 12). In terms of other influencing factors, the road recovery clusters in Miyagi prefecture relate only to important roads (Table 13).

The topography of Fukushima prefecture is characterized by a larger number of high mountains, while Miyagi prefecture has more plains. The correlation between road restoration patterns and road elevation in both prefectures generally corresponds to the prefectures’ topographic features. This also shows that regardless of the damage, important roads still influence the speed of road recovery.

## 5. Conclusions

Using cluster analysis, we divided the road recovery in Fukushima prefecture after the earthquake into seven clusters. The results of the cluster analysis were validated using discriminant analysis. We represented the clustering results on the map and observed that road recovery patterns were correlated with topography and road importance. Their correlation in Fukushima and Miyagi prefectures was verified by objective data.

Previous studies [17,18] have not performed outcome testing, which was performed in this study. The results of this study both validated the conclusions of previous studies [17,18] suggesting a correlation between road recovery and topography, while overturning the conclusions of a previous study [18] correlating road recovery and population density. In addition, a new factor related to road restoration was explored, namely road importance. For future exploration of other factors affecting road recovery, the research framework is presented to provide a methodological basis.

This study applied cluster analysis to determine the patterns of road recovery after the earthquake in Fukushima prefecture’s municipalities. In addition to the features of topography and road importance, we want to identify more features that will reflect the recovery of the municipal roads. In the future, we hope to use the characteristics of road recovery derived from previous disaster studies to predict road recovery in municipalities that may experience earthquakes.

## Figures and Tables

**Figure 1 sensors-22-02213-f001:**
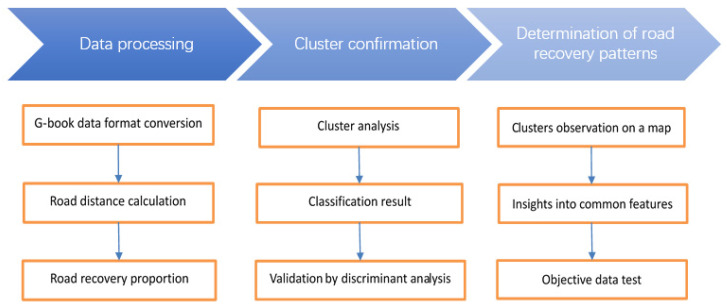
Research flowchart for determining road recovery patterns.

**Figure 2 sensors-22-02213-f002:**
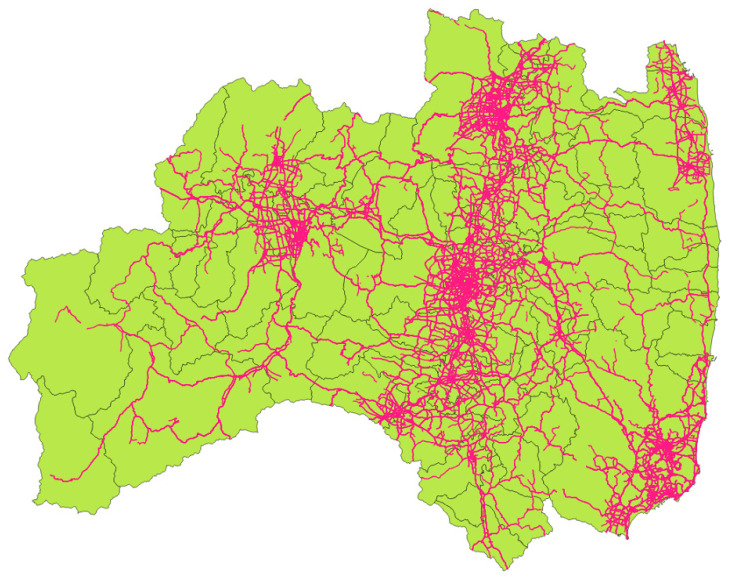
Vehicle tracking map of Fukushima prefecture on 30 September 2011.

**Figure 3 sensors-22-02213-f003:**
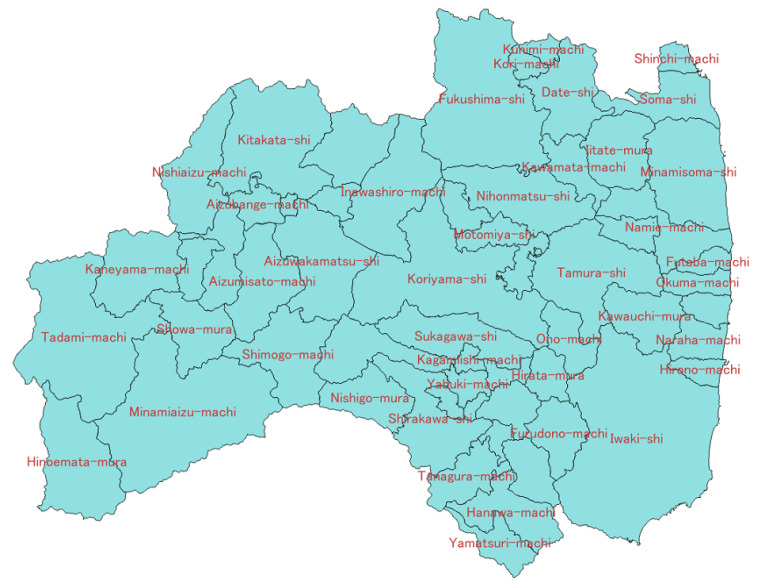
Municipalities of Fukushima prefecture.

**Figure 4 sensors-22-02213-f004:**
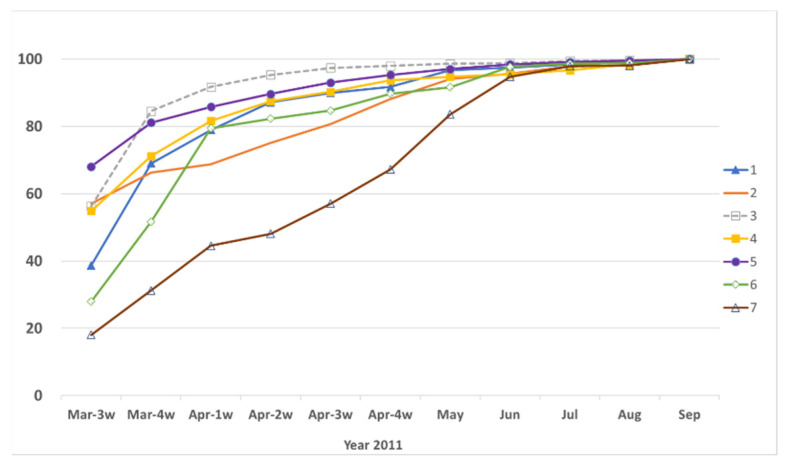
Road recovery conditions of the seven clusters in Fukushima Prefecture.

**Figure 5 sensors-22-02213-f005:**
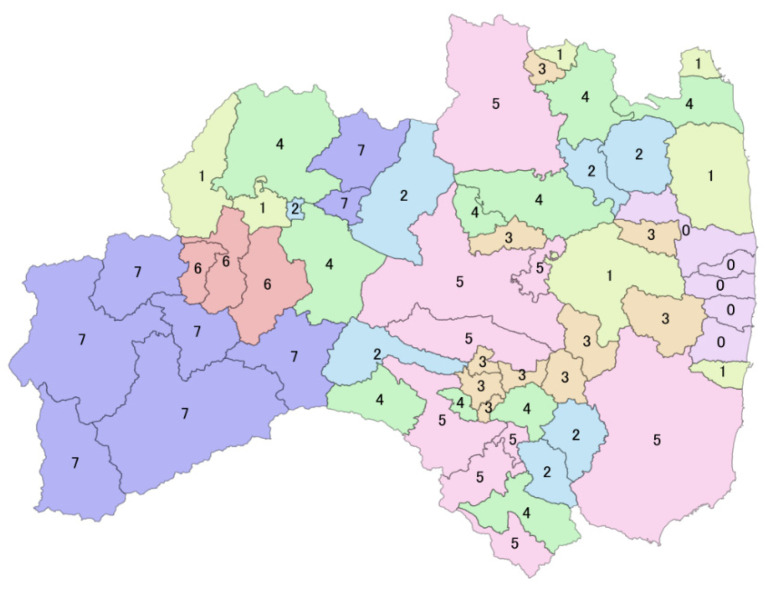
Municipalities with similar road recovery speeds were divided into seven clusters.

**Figure 6 sensors-22-02213-f006:**
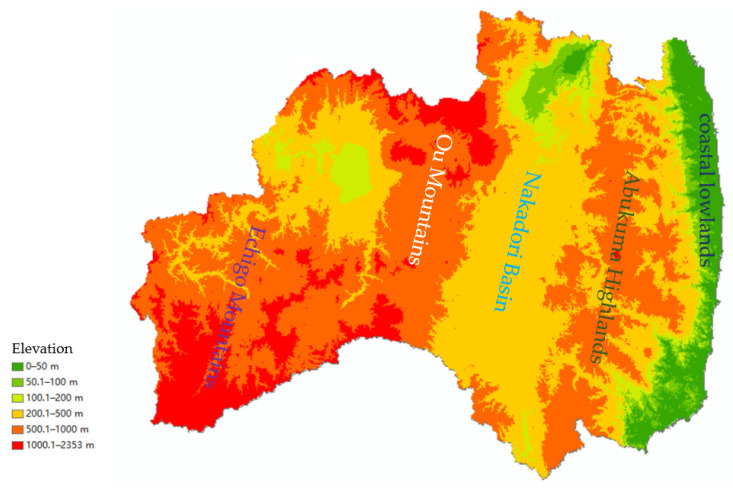
Topographical map of Fukushima prefecture.

**Figure 7 sensors-22-02213-f007:**
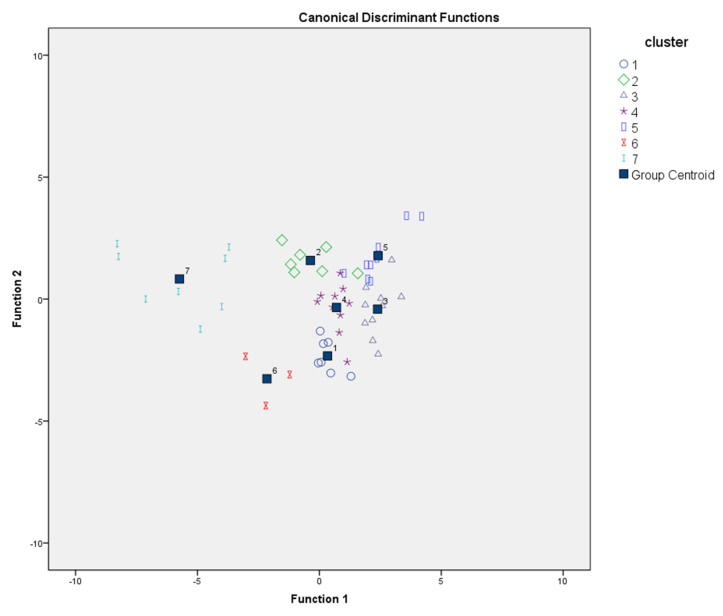
Canonical discriminant functions.

**Table 1 sensors-22-02213-t001:** Seven clusters of municipalities with similar road recoveries.

Cluster	Municipality	Mar.—3 w	Mar.—4 w	Apr.—1 w	Apr.—2 w	Apr.—3 w	Apr.—4 w	May	Jun.	Jul.	Aug.	Sep.
1	Hirono-machi	38	69	71	88	92	93	97	98	99	100	100
1	Minamisoma-shi	34	71	83	89	93	93	95	97	98	99	100
1	Shinchi-machi	44	64	77	86	88	94	97	98	98	99	100
1	Kunimi-machi	33	71	79	88	89	89	98	98	98	98	100
1	Tamura-shi	43	71	80	87	91	94	96	96	98	99	100
1	Aizubange-machi	41	69	81	85	89	92	96	97	97	100	100
1	Nishiaizu-machi	39	67	81	88	88	88	99	100	100	100	100
2	Iitate-mura	55	63	67	76	82	86	93	93	100	100	100
2	Kawamata-machi	56	71	71	76	85	90	95	96	98	100	100
2	Tenei-mura	49	63	72	76	80	85	97	97	98	99	100
2	Furudono-machi	57	58	59	73	86	93	93	93	99	100	100
2	Samegawa-mura	72	72	73	76	78	95	98	98	98	100	100
2	Inawashiro-machi	52	68	71	72	77	85	95	97	98	100	100
2	Yugawa-mura	59	68	69	78	78	83	89	96	96	98	100
3	Katsurao-mura	60	74	96	100	100	100	100	100	100	100	100
3	Kawauchi-mura	40	95	98	98	99	99	99	99	100	100	100
3	Kori-machi	57	86	93	95	96	96	100	100	100	100	100
3	Kagamiishi-machi	57	85	89	95	98	98	98	98	99	100	100
3	Motomiya-shi	69	87	93	95	97	98	99	99	100	100	100
3	Ono-machi	51	88	95	95	97	97	98	98	98	98	100
3	Hirata-mura	57	82	90	95	96	96	97	97	100	100	100
3	Nakajima-mura	58	81	88	92	97	97	97	97	98	98	100
3	Tamakawa-mura	52	78	85	93	98	100	100	100	100	100	100
3	Yabuki-machi	63	90	91	94	96	98	98	100	100	100	100
4	Soma-shi	54	77	84	86	92	95	96	96	96	99	100
4	Date-shi	55	71	81	88	93	95	96	97	98	100	100
4	Nihonmatsu-shi	59	75	80	84	87	91	94	95	97	99	100
4	Otama-mura	46	75	84	89	91	93	94	94	94	100	100
4	Hanawa-machi	60	65	75	85	87	88	88	88	88	88	100
4	Ishikawa-machi	57	69	89	92	93	97	97	97	100	100	100
4	Izumizaki-mura	55	78	87	89	91	95	96	97	99	100	100
4	Nishigo-mura	53	71	82	91	92	99	100	100	100	100	100
4	Aizuwakamatsu-shi	55	69	80	83	86	92	95	97	98	100	100
4	Kitakata-shi	55	61	74	86	91	92	94	95	97	98	100
5	Iwaki-shi	60	81	88	90	93	95	97	98	99	99	100
5	Fukushima-shi	66	80	85	88	90	95	96	98	99	99	100
5	Koriyama-shi	66	84	89	91	94	96	97	98	99	99	100
5	Miharu-machi	60	80	83	85	90	97	99	99	100	100	100
5	Sukagawa-shi	64	80	85	90	92	95	97	97	98	100	100
5	Asakawa-machi	67	76	79	86	95	98	99	99	99	99	100
5	Shirakawa-shi	67	80	86	89	92	92	97	98	99	99	100
5	Tanagura-machi	82	86	89	94	98	98	99	99	100	100	100
5	Yamatsuri-machi	81	83	90	93	93	93	93	100	100	100	100
6	Aizumisato-machi	39	51	71	75	81	93	93	96	98	99	100
6	Mishima-machi	16	54	80	80	82	82	84	98	98	98	100
6	Yanaizu-machi	29	50	87	92	92	94	98	100	100	100	100
7	Bandai-machi	28	52	56	59	59	70	95	97	98	98	100
7	Kaneyama-machi	0	13	61	61	64	64	65	97	100	100	100
7	Kitashiobara-mura	39	44	48	54	71	82	97	98	98	98	100
7	Showa-mura	0	33	46	47	70	73	95	95	95	95	100
7	Hinoemata-mura	0	0	0	0	0	31	61	100	100	100	100
7	Minamiaizu-machi	32	47	59	64	64	77	84	88	99	99	100
7	Shimogo-machi	41	57	59	62	75	85	93	94	99	99	100
7	Tadami-machi	4	4	27	38	53	55	78	90	94	95	100

**Table 2 sensors-22-02213-t002:** Eigenvalues.

Function	Eigenvalue	% of Variance	Cumulative %	Canonical Correlation
1	8.366 ^a^	65.6	65.6	0.945
2	2.648 ^a^	20.8	86.3	0.852
3	1.002 ^a^	7.9	94.2	0.707
4	0.630 ^a^	4.9	99.1	0.622
5	0.074 ^a^	0.6	99.7	0.262
6	0.042 ^a^	0.3	100.0	0.202

^a^ The first 6 canonical discriminant functions were used in the analysis.

**Table 3 sensors-22-02213-t003:** Wilks’ lambda.

Test of Function(s)	Wilks’ Lambda	Chi-Square	df	Sig.
1	0.008	214.775	60	0.000
2	0.075	115.227	45	0.000
3	0.274	57.634	32	0.004
4	0.548	26.743	21	0.180
5	0.894	5.010	12	0.958
6	0.959	1.851	5	0.869

**Table 4 sensors-22-02213-t004:** The number of road recovery percentages in seven clusters of Fukushima.

Cluster	Mar.—3 w	Mar.—4 w	Apr.—1 w	Apr.—2 w	Apr.—3 w	Apr.—4 w	May	Jun.	Jul.	Aug.	Sep.
1	39	69	79	87	**90**	92	97	98	98	99	100
2	57	66	69	75	81	88	**94**	96	98	100	100
3	56	85	**92**	95	97	98	99	99	99	100	100
4	55	71	82	87	**90**	94	95	96	97	98	100
5	68	81	86	**90**	93	95	97	98	99	100	100
6	28	52	79	82	85	**90**	92	98	99	99	100
7	18	31	45	48	57	67	84	**95**	98	98	100

**Table 5 sensors-22-02213-t005:** Standardized canonical discriminant function coefficients.

Independent Variable	Function	
	1	2
Mar.—3 w	0.377	1.30
Mar.—4 w	1.00	0.127
Apr.—1 w	−1.12	1.20
Apr.—2 w	1.29	−2.85
Apr.—3 w	0.607	2.54
Apr.—4 w	−0.396	−2.00
May	−0.870	0.055
June	0.836	−0.402
July	−0.374	0.333
August	−0.003	0.228

**Table 6 sensors-22-02213-t006:** Structure matrix.

Independent Variable	Function	
	1	2
Mar.—3 w	0.569 *	0.433
Mar.—4 w	0.664 *	0.010
Apr.—1 w	0.576 *	−0.260
Apr.—2 w	0.617 *	−0.314
Apr.—3 w	0.477 *	−0.208
Apr.—4 w	0.476 *	−0.171
May	0.287	−0.082
June	0.180	−0.084
July	0.067	0.014
August	0.086	0.030

*. Largest absolute correlation between each variable and any discriminant function.

**Table 7 sensors-22-02213-t007:** Canonical discriminant function coefficients.

Independent Variable		Function	
		1	2
Mar.—3 w	X_1_	0.039	0.136
Mar.—4 w	X_2_	0.104	0.013
Apr.—1 w	X_3_	−0.123	0.131
Apr.—2 w	X_4_	0.148	−0.329
Apr.—3 w	X_5_	0.063	0.263
Apr.—4 w	X_6_	−0.053	−0.270
May	X_7_	−0.144	0.009
June	X_8_	0.356	−0.171
July	X_9_	−0.190	0.169
August	X_10_	−0.001	0.119
(Constant)		−14.3	−1.76

**Table 8 sensors-22-02213-t008:** Functions at group centroids.

Cluster	Function	
	1	2
1	0.336	−2.33
2	−0.367	1.58
3	2.39	−0.412
4	0.702	−0.346
5	2.41	1.78
6	−2.15	−3.27
7	−5.74	0.824

**Table 9 sensors-22-02213-t009:** Classification results.

Cluster			Predicted Group Membership							Total
			1	2	3	4	5	6	7	
Original	Count	1	6	0	0	1	0	0	0	7
		2	0	6	0	1	0	0	0	7
		3	0	0	9	0	1	0	0	10
		4	1	0	0	9	0	0	0	10
		5	0	0	0	0	9	0	0	9
		6	0	0	0	0	0	3	0	3
		7	0	0	0	0	0	0	8	8
	%	1	85.7	0.0	0.0	14.3	0.0	0.0	0.0	100.0
		2	0.0	85.7	0.0	14.3	0.0	0.0	0.0	100.0
		3	0.0	0.0	90.0	0.0	10.0	0.0	0.0	100.0
		4	10.0	0.0	0.0	90.0	0.0	0.0	0.0	100.0
		5	0.0	0.0	0.0	0.0	100.0	0.0	0.0	100.0
		6	0.0	0.0	0.0	0.0	0.0	100.0	0.0	100.0
		7	0.0	0.0	0.0	0.0	0.0	0.0	100.0	100.0

92.6% of original grouped cases correctly classified.

**Table 10 sensors-22-02213-t010:** Correlations between road restoration clusters and road elevation in Fukushima prefecture.

Cluster	Elevations				
	<50 m	50–100 m	100–200 m	200–500 m	>500 m
Pearson Correlation	−0.265	−0.232	−0.291	0.150	0.311
Sig.	0.053	0.091	0.033	0.279	0.022

**Table 11 sensors-22-02213-t011:** Correlations between road restoration clusters and affected factors in Fukushima prefecture.

Cluster	Road Importance	Road Density	Population Density	Measured Seismic Intensities	Road Closure Rate Due to Snow
Pearson Correlation	0.417	−0.416	−0.190	−0.637	0.729
Sig.	0.002	0.002	0.169	0.000	0.000

**Table 12 sensors-22-02213-t012:** Correlations between road restoration clusters and road elevation in Miyagi prefecture.

Cluster	Elevations				
	<50 m	50–100 m	100–200 m	200–500 m	>500 m
Pearson Correlation	−0.370	0.129	0.222	0.402	0.204
Sig.	0.020	0.435	0.174	0.011	0.214

**Table 13 sensors-22-02213-t013:** Correlations between road restoration clusters and affected factors in Miyagi prefecture.

Cluster	Road Importance	Road Density	Population Density	Measured Seismic Intensities	Road Closure Rate Due to Snow
Pearson Correlation	0.365	−0.242	−0.163	−0.081	0.145
Sig.	0.022	0.137	0.321	0.624	0.380

## Data Availability

Data sharing is not applicable to this article.

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
