# Peer review of "Cluster Analysis and Discriminant Analysis for Determining Post-Earthquake Road Recovery Patternsâ€"

_sensors, 2022, doi:10.3390/s22062213_

Round 1
Reviewer 1 Report
The manuscript is more like a data analysis report rather than a serious research paper. The authors applied the cluster analysis to analyze the data of vehicle driving in Fukushima prefecture to classify the road recovery conditions. Yet, why such cluster can express the characteristics of the road recovery conditions is not fully discussed. The manuscript provides the analysis tables and maps, but not provides sufficient discussion on the ideas behind and the scientific evidence revealed. Therefore, the manuscript is not suitable to be published at current form. Here are some of my suggestions:
- The author can provide a flowchart to express the design of their methodology. It is better to provide detailed explains why such methodology framework works.
- Provide more detailed discussion on the analysis results that related to the road recovery conditions. It is not a good way that only discuss the analysis results of only the data and tables exported from the statistical software.
- Discuss more the contribution of your work. It is better to have performance experiment or comparation with other related works.
Author Response
Dear reviewer,
Thank you for your comments. I think your comments are spot on. I revised my paper in response to your comments.
For your suggestions, I did the revisions as follows:
1. The author can provide a flowchart to express the design of their methodology. It is better to provide detailed explains why such methodology framework works.
Response 1: I made a flowchart about our research. Flowcharts are indeed very useful. My thinking becomes much clearer as well. With this, I will continue this step in all my future papers.
2.Provide more detailed discussion on the analysis results that related to the road recovery conditions. It is not a good way that only discuss the analysis results of only the data and tables exported from the statistical software.
Response 2: I rewrite the discussion about cluster analysis results. I originally intended to express the results through objective data analysis, which I obviously did not do well.
3. Discuss more the contribution of your work. It is better to have performance experiment or comparation with other related works.
Response 3: In the introduction section, I rewrite the related work. The shortcomings of the previous study, namely the cluster analysis results were not validated and observation of the cluster from a map was not tested by objective data. By comparing my research with previous studies, highlighting the research significance of my study.
I am glad you gave a review this time and know from your comments my shortcomings. I look forward to your comments.
Best regards,
Jieling Wu
Reviewer 2 Report
This research divided the road recovery in Fukushima prefecture after the earthquake into seven cluster. By the cluster analysis, the results are validated by discriminant analysis.
In addition, the research representes the clustering results on the map and observed that road recovery patterns are correlated with topography. Although the practical meaning of this research is enough, the scientific significance is lack. So, the authors should explore the deep factors and patterns of road recovery after the earthquake.
The detail comments for authors as follows:
(1) Firstly, the abstract is so simple, so the authors should conclude it again.
(2) Secondly, the source of data is not clear. Besides, the role of data is not introduced in detail. Such as Vehicle Tracking Map and Digital Road Map, I can't see what role they play in this research.
(3) Thirdly, this research just shows the cluster analysis results, but the cluster analysis method is not introduced.
(4) The most obvious problem is that this research is not logical and integral, so I suggest that this manuscript should be modified and supplemented.
Author Response
Dear reviewer,
Thank you for your serious comments. I think your comments are spot on. Apart from the topography, I discussed deeply road recovery patterns and made modifications to explore other factors that affect the recovery of the road.
For your comments, I did the revisions as follows:
(1) Firstly, the abstract is so simple, so the authors should conclude it again.
Response 1: I rewrite the abstract. I described the shortcomings of previous studies and the contribution of our work.
(2) Secondly, the source of data is not clear. Besides, the role of data is not introduced in detail. Such as Vehicle Tracking Map and Digital Road Map, I can't see what role they play in this research.
Response 2: Regarding the data, I really didn't present it clearly before. I reintroduced it in my paper. The cluster was derived by collating vehicle tracking map. And digital road map is used to test the observations.
(3) Thirdly, this research just shows the cluster analysis results, but the cluster analysis method is not introduced.
Response 3: You are absolutely right. I have reintroduced cluster analysis.
(4) The most obvious problem is that this research is not logical and integral, so I suggest that this manuscript should be modified and supplemented.
Response 4: Regarding logic and integral, I made a flow chart so that the flow of the study would be clearer. Attention was also paid to the connection between paragraphs, and the order of some paragraphs was adjusted.
I am glad you gave a review this time and know from your comments my shortcomings. I look forward to your comments.
Best regards,
Jieling Wu
Round 2
Reviewer 1 Report
The manuscript has largely been improved. I only have some minor comments.
- It is better to add more details in the method section. For example, the cluster analysis. There are different methods in the cluster analysis, which may produce different results. The choice of distance and the way how to compute clusters from the distance matrics will largely affect the cluster and canonical discriminant analysis results. Therefore, adding more details, e.g. adding some equations, in the method section, will lead to a clearer presentation.
- I still found the explanation of different clusters is weak. As the readers may not be familiar with the geographical situations of the research area, they will be confused when reading section 3.1. I suggest the authors add more explanations.
- For the whole paper, I suggest the authors add more discussion on the topic of "sensors". Otherwise, it is more technical rather than scientific.
Author Response
- It is better to add more details in the method section. For example, the cluster analysis. There are different methods in the cluster analysis, which may produce different results. The choice of distance and the way how to compute clusters from the distance matrics will largely affect the cluster and canonical discriminant analysis results. Therefore, adding more details, e.g. adding some equations, in the method section, will lead to a clearer presentation.
Response 1: Thank you for your advice. From the full text, the cluster analysis is confirmed to be somewhat short. Following your comments, I have added the choice of method, the distance equation, and a description of the choice of the number of clusters. As I previously thought that cluster analysis was a relatively mature analysis method and that the results could be obtained by importing directly into SPSS, I did not give much thought to writing a description of the cluster analysis method.
- I still found the explanation of different clusters is weak. As the readers may not be familiar with the geographical situations of the research area, they will be confused when reading section 3.1. I suggest the authors add more explanations.
Response 2: I have added a topographical map of Fukushima prefecture so that it will look clearer. And I have also added a definition of what type of features the municipalities in each cluster belong to.
- For the whole paper, I suggest the authors add more discussion on the topic of "sensors". Otherwise, it is more technical rather than scientific.
Response 3: I have added additional discussion in section 3.2.1, section 4.2, and section 5 to highlight the contribution of this study.
Reviewer 2 Report
The manuscript has been modified by the comments, but there still be some problems, as follows,
(1) The abstract has been modified, but the contribution of this research needs to be highlighted.
(2) The data source has been introduced clearly, and data processing has been added.
(3) This research need introduce the cluster analysis method and process.
(4) In order to make the article be more logical, the deep factors and patterns of road recovery after the earthquake need be explored.
(5) The most important is that the authors need further conclude the scientific significance of this research.
Author Response
(1) The abstract has been modified, but the contribution of this research needs to be highlighted.
Response 1: Thank you for your kind advice again. I reiterated the results of the cluster analysis and relevant factors affecting road recovery from previous studies were not validated. And it does so in this study, with a research framework presented as a contribution.
(2) The data source has been introduced clearly, and data processing has been added.
Response 2: Yes. This time I have also added a figure of Vehicle Tracking Map.
(3) This research needs introduce the cluster analysis method and process.
Response 3: Yes. I added the cluster analysis method and process in section 2.4. Following your comments, I have added the choice of method, the distance equation, and a description of the choice of the number of clusters. As I previously thought that cluster analysis was a relatively mature analysis method and that the results could be obtained by importing directly into SPSS, I did not give much thought to writing a description of the cluster analysis method.
(4) In order to make the article be more logical, the deep factors and patterns of road recovery after the earthquake need be explored.
Response 4: Yes. Both times you mentioned this, my last revision in 4.1 added 5 additional impact factors and analyzed them with objective data. Was that not enough or did I not understand what you meant. This time I added a topographical map of Fukushima Prefecture so that it will look clearer. And I have also added a definition of what type of features the municipalities in each cluster belong to.
(5) The most important is that the authors need further conclude the scientific significance of this research.
Response 5: I added additional discussion in section 3.2.1, section 4.2, and section 5 to highlight the contribution of this study.